# Design and Synthesis of New Anthranyl Phenylhydrazides: Antileishmanial Activity and Structure–Activity Relationship

**DOI:** 10.3390/ph16081120

**Published:** 2023-08-09

**Authors:** Claudia do Carmo Maquiaveli, Edson Roberto da Silva, Barbara Hild de Jesus, Caio Eduardo Oliveira Monteiro, Tiago Rodrigues Navarro, Luiz Octavio Pereira Branco, Isabela Souza dos Santos, Nanashara Figueiredo Reis, Arieli Bernardo Portugal, João Luiz Mendes Wanderley, André Borges Farias, Nelilma Correia Romeiro, Evanoel Crizanto de Lima

**Affiliations:** 1Laboratório de Farmacologia e Bioquímica (LFBq), Departamento de Medicina Veterinária, Universidade de São Paulo Faculdade de Zootecnia e Engenharia de Alimentos, Av. Duque de Caxias Norte 225, Pirassununga 13635-900, SP, Brazil; 2Laboratório de Catálise e Síntese de Substâncias Bioativas, Instituto Multidisciplinar de Química, CM UFRJ-Macaé, Universidade Federal do Rio de Janeiro, Macaé CEP 27971-525, RJ, Brazil; 3Laboratório de Imunoparasitologia, Instituto de Ciências Médicas, Centro Multidisciplinar UFRJ, Macaé CEP 27979-000, RJ, Brazil; 4Programa de Pós Graduação em Biociências e Biotecnologia, Universidade Estadual do Norte Fluminense, Campos dos Goytacazes CEP 28013-602, RJ, Brazil; 5Unidad Académica de Yucatán, Instituto de Investigaciones en Matemáticas Aplicadas y en Sistemas, Universidad Nacional Autónoma de México, Mérida 97302, Yucatán, Mexico; 6Integrated Laboratory of Scientific Computing (LICC), Federal University of Rio de Janeiro (UFRJ)—Campus Macaé, Aluízio Silva Gomes Avenue 50, Granjas Cavaleiros, Macaé 27930-560, RJ, Brazil

**Keywords:** *Leishmania*, polyamines, arginase, PTR1, anthranyl hydrazides

## Abstract

Leishmaniasis is a neglected tropical disease affecting millions of people worldwide. A centenary approach to antimonial-based drugs was first initiated with the synthesis of urea stibamine by Upendranath Brahmachari in 1922. The need for new drug development led to resistance toward antimoniates. New drug development to treat leishmaniasis is urgently needed. In this way, searching for new substances with antileishmanial activity, we synthesized ten anthranyl phenylhydrazide and three quinazolinone derivatives and evaluated them against promastigotes and the intracellular amastigotes of *Leishmania amazonensis*. Three compounds showed good activity against promastigotes 1b, 1d, and 1g, with IC_50_ between 1 and 5 μM. These new phenylhydrazides were tested against *Leishmania* arginase, but they all failed to inhibit this parasite enzyme, as we have shown in a previous study. To explain the possible mechanism of action, we proposed the enzyme PTR1 as a new target for these compounds based on in silico analysis. In conclusion, the new anthranyl hydrazide derivatives can be a promising scaffold for developing new substances against the protozoa parasite.

## 1. Introduction

Leishmaniasis is a neglected tropical disease that affects 12 million people in the world; it is found in 98 countries, and an estimated one billion people are at risk of the disease [1]. The treatments for leishmaniasis disease are based on antimonial pentavalent as the first choice. This centenary approach-based drug was initiated first with the synthesis of urea stibamine by Upendranath Brahmachari in 1922 [2]. The urea stibamine significantly impacts the treatment of leishmaniasis, but the high toxicity led to the development of sodium stibogluconate in 1950 and N-methylglucamine antimonate (glucantime). However, antimonials have several limitations, including high toxicity, long treatment duration, and emerging resistance. This has prompted the exploration of alternative treatment options for leishmaniasis [3].

While antimonials have been the mainstay of leishmaniasis treatment for a century, alternative drugs such as miltefosine, amphotericin B, and pentamidine have emerged as effective options [1]. Amphotericin B, a polyene antifungal drug, interferes with membrane ergosterol, while pentamidine is involved in the polyamine biosynthesis pathway. These drugs are also highly toxic and unavailable for oral delivery [4].

Miltefosine is the first oral drug approved for leishmaniasis treatment. Miltefosine has shown promising efficacy against various forms of the disease, including visceral leishmaniasis (VL) and cutaneous leishmaniasis (CL) [5]. Despite being active against antimonial-resistant strains [5], resistance was described [6]. In Brazil, because of resistance to the miltefosine treatment in human visceral leishmaniasis, the drug is approved only to treat dogs infected by *Leishmania* [7].

Researchers continue to explore old and new targets for rational drug design. For example, a second line of alternative treatment was carried out using allopurinol, which blocks the de novo synthesis of purines [8,9,10]. A world effort to develop new drugs against this neglected tropical disease involves low private and public investments, mainly at the academic level. Repurposing was also explored with marbofloxacin, a fluoroquinolone that inhibits the topoisomerase (DNA gyrase) and tamoxifen, an antitumoral agent [11,12].

The polyamine biosynthesis pathway [13] is a promising route to drug development targeting arginase [14] and ornithine decarboxylase (ODC) [15]. We previously explored the synthesis of several scaffolds, such as cinnamic acid, pyrazolopyrimidines, and phenylhydrazides, using *Leishmania* arginase as a target. They all showed selective arginase inhibition and good activity against parasites, inspiring a new scaffold of anthranyl phenylhydrazides [16]. Until now, no other hydrazides with leishmanicidal activity have been described in the literature, except for those previously published by our group [16], for which its structure was based on eflornithine, an inhibitor of the ODC enzyme. This study tests 13 structural hydrazides related to the previously prepared α,α-difluoromethylhydrazide as a prototype for designing anthranyl phenylhydrazides using molecular simplification. These anthranyl phenylhydrazides showed better activity against *Leishmania amazonensis* promastigotes than the previous α,α-difluoromethylhydrazide series [16].

## 2. Results and Discussion

### 2.1. Chemistry

Anthranyl hydrazides were carried out with isatoic anhydride and phenylhydrazine under reflux in ethanol for 2 h [17]. In cases where phenylhydrazine is available in salt form, NaOH was used to obtain the free nucleophile. Posteriorly, hydrazides 1a, 1d, and 1g were cyclized, furnishing respective quinazolin-4(3H)-ones 2a, 2b, and 2c using acetyl chloride and CH_3_CN as the solvent at room temperature for 3 h. The reaction with acetic anhydride was also conducted, but the reaction times were higher, ~6–9 h.

### 2.2. Planning of Anthranyl Hydrazides Series

Based on our previous results with the α-α-difluorophenyl hydrazide series, new compounds were designed from the phenylhydrazide previously characterized as a pharmacophoric group for antileishmanial activity [16]. Figure 1 shows the phenylhydrazide and aminophenyl groups in the compounds using a molecular simplification approach.

### 2.3. Activity in Promastigotes and Intracellular Amastigotes

The study of α-α-difluorophenyl hydrazides compounds revealed the phenylhydrazide group as a new pharmacophoric group against *L. amazonensis* [16]. Another series of anthranyl hydrazide compounds were planned as potential agents against Leishmania. In a screening of *L. amazonensis* promastigote cultures, all compounds showed inhibition above 50% at 100 μM. We screened at 10 μM and selected four compounds to determine the IC_50_. However, **1e** showed myelotoxicity in an animal model [17], and we did not determine the IC_50_. The best result in the trial at 10 μM was observed for compound **1g**, which showed 97% growth inhibition with an IC_50_ of 1.3 μM (Table 1).

Thus, we determined the IC_50_ for compounds **1b**, **1d,** and **1g** (Figure 2). Compound **1g** was the most potent, with an IC_50_ of 1.3 μM. An ANOVA test was applied, followed by Tukey’s post hoc test, which indicated that **1d** differs significantly from **1b** (*p* < 0.05) and **1g** differs from **1d** and **1b** (*p* < 0.001).

Despite the good activity against *L. amazonensis* promastigotes, the compounds failed against intracellular amastigotes using peritoneal murine macrophage (data not shown). *Leishmania* spp. use several approaches for resistance against drugs [3]. One is the ABC transporter MRPA (PGPA) in antimony resistance described for *Leishmania infantum* [18]. The PRP1, an ABC transporter superfamily member, was correlated with pentamidine resistance in *Leishmania major* [19]. A new study must be performed to understand why the hydrazide compounds were inactive against intracellular amastigotes.

### 2.4. Structure–Activity Relationship Analysis of the Anthranyl Hydrazides Series

By comparing the IC_50_ in promastigote culture, it can be inferred that halogenation with chlorine or bromine in the p-phenyl (R4) position increases antileishmanial activity. The leishmanial inhibition growth was also observed with bromine at position R1 (compound **1b**, IC_50_ = 4.2 μM), and higher inhibition was observed with bromine at position R1 and R4 in compound **1g** (IC_50_ = 1.3 μM).

The promastigote culture test results also showed that the anthranyl hydrazide series was superior to the α-α-difluorohydrazides compounds. With the culture result obtained for the α-α-difluorohydrazides [16] compound series, we identified the pharmacophoric phenylhydrazide group and produced a new series of compounds. In the present study, the insertion of bromine into the phenyl groups at positions R1 and R4 (Figure 3) increased the growth-inhibitory potency of the promastigote forms by about 10-fold. The new compound **1g** showed an IC_50_ = 1.3 μM, while compound **4** showed an IC_50_ of 12.7 μM [16].

The derived compounds **2a**, **2b**, and **2c** were inactive, showing that the scaffold used in compounds **1(a–j)** is essential for the activity against *L. amazonensis*.

### 2.5. Mechanism of Antileishmanial Activity

Based on previous results with phenylhydrazides [16] that inhibit *Leishmania* arginase, we tested the possible enzyme inhibition in vitro. All anthranyl phenylhydrazides failed as *Leishmania* arginase inhibitors, and to propose a new method of parasite killing, we performed target fishing as follows.

### 2.6. Target Fishing for Anthranyl Phenylhydrazides

A target-fishing approach was performed using chemogenomics to provide new targets for phenylhydrazide inhibitors. Several studies have demonstrated the ability of this approach to predict promising targets [20,21]. Thus, some targets, including arginase, were predicted as potential targets for inhibitors (Appendix A). Unfortunately, the results in this work revealed that the synthesized inhibitors had no activity on arginase, leading to the search for new targets for *L. amazonensis*. Due to the absence or lack of information about *L. amazonensis* in the database, it was impossible to directly predict a target. However, the target-fishing predictions by chemogenomics suggested pteridine reductase 1 (PTR1) from *L. major* as a possible target. An alignment (Appendix A) was performed on the primary sequence of PTR1 to search for a homologous protein for *L. amazonensis*, indicating aldoketoreductase (GenBank AAB61214.1) with 100% query cover and 91.67% identity.

Pteridine reductase is an NADPH-dependent reductase that catalyzes the reduction in folate and biopterin into their biologically active forms, tetrahydrofolate and tetrahydrobiopterin, respectively [22]. It is the only enzyme to perform this reduction in *Leishmania*, proving essential for their in vivo growth by gene knockout studies [23]. Furthermore, *Leishmania* is auxotrophic for folate and other pterins required in critical pathways related to nucleic acids and protein synthesis [24,25]. Therefore, the drug development of new inhibitors for PTR1 would be a promising strategy for the treatment of leishmaniasis [26,27,28]. Although target fishing provides significant results on possible molecular targets, this technique depends on molecular scaffolds in the database, often with sparse data for specific organisms. Thus, molecular docking was performed to obtain the theoretical binding mode of the designed compounds with PTR1 from *L. amazonensis*. Despite the PTR1 binding site from *L. amazonensis* being very similar to *L. major* (RMSD of 1.30 Å over 264 residues), 24 amino acids are not conserved (Appendix A). Of these 24 residues, only two, A229 and Y240, are within 5 Å of the inhibitor methotrexate. The catalytic center is formed by β1, β4, β5, and β6, located in the C-terminal regions, α1 and α5 of the N-terminal, and a loop between β6 and α6 [29]. The results showed that compound **1g** could interact with two residues of the catalytic triad (Y193 and D180) by hydrogen bonding. Besides that, π–π stacked interactions with F113 and the cofactor were also observed (Figure 4). Notably, the other molecules, in general, presented a similar putative binding mode (Appendix A), indicating that this molecular scaffold may be relevant to the drug design of new candidates to treat leishmaniasis. Similarity with the crystallized co-inhibitor methotrexate was also observed (Appendix A), mainly concerning the interaction with the catalytic residue Y193 (Y194 in PTR1 from *L. major*). Gourley et al. described PTR1’s mechanism of action, involving hydrogen bonds between Y194 and a nucleophilic center of the substrate, bound to stabilize the transition state to provide the proton transferred in the reaction [29]. Therefore, the observed interactions support the hypothesis of PTR1 as a target of the studied compounds.

Further studies need to be conducted to validate the molecular docking hypothesis into PTR1 inhibition, such as enzymatic binding assays. However, PTR1 is a recognized target for leishmanicidal drug design [26,30] since it provides enough folate to guarantee the parasite’s survival [31]. Taken together, this preliminary study, associated with the inhibition in promastigote cells and the prediction by chemogenomics, is a promising molecular target for developing new inhibitors based on the phenylhydrazide scaffold.

## 3. Materials and Methods

### 3.1. Chemicals

The M199 medium was supplemented with 10% fetal bovine serum, 5 ppm of hemin, 50 μg/mL streptomycin, and 100 U penicillin. Hanks Balanced Salt Solution (GIBCO^®^ HBSS) was purchased from Thermo Life technologies, and MTT 3-(4,5-dimethylthiazol-2-yl)-2,5-diphenyl tetrazolium bromide was purchased from Sigma.

### 3.2. Synthesis

The anthranyl hydrazides were prepared using the experimental procedure previously reported, in which improvements were made compared to the literature [17]. Quinazolin-4(*3H*)-ones **2a**, **2b,** and **2c** were prepared from respective phenyl hydrazides (1.89 mmol), using acetyl chloride (9.45 mmol) and CH_3_CN (2.5 mL) as solvent at room temperature. TLC accomplished reactions by 3 h, and after this period, cold water was added to the reaction mixture. The solid was filtered, having sufficient purity to be tested posteriorly.

#### 3.2.1. Synthesis of 2-amino-5-bromo-N′-phenylbenzohydrazide (**1b**)

Yellow solid, 55% yield. GC-MS: *m*/*z* 307 (34%), *m*/*z* 305 (34%), *m*/*z* 200 (100%), *m*/*z* 198 (99%), *m*/*z* 170 (21%). ESI-MS(+) C_13_H_12_BrN_3_NaO: *m*/*z* 328.0055.

#### 3.2.2. Synthesis of 2-amino-N′-(2-bromophenyl)benzohydrazide (**1c**)

Pale solid, 95% yield. GC-MS: *m*/*z* 307 (15%), *m*/*z* 305 (15%), *m*/*z* 120 (100%), *m*/*z* 92 (22%). ESI-MS(+) C_13_H_13_BrN_3_O: *m*/*z* 306.0237; C_13_H_12_BrN_3_NaO: *m*/*z* 328.0053.

NMR ^1^H (MeOD, 400 MHz, δ): 7.62 (d, 1H, *J* = 7.9 Hz), 7.44 (d, 1H, *J* = 7.9 Hz), 7.25 (t, 1H, *J* = 8 Hz), 7.21 (t, 1H, *J* = 8 Hz), 6.94 (d, 1H, *J* = 8.2 Hz), 6.78 (d, 1H, *J* = 8.3 Hz), 6.73 (t, 1H, *J* = 7.8 Hz), 6.67 (t, 1H, *J* = 7.8 Hz).

#### 3.2.3. Synthesis of 2-amino-5-bromo-N′-(4-bromophenyl) benzohydrazide (**1g**)

Yellow solid, 52% yield. GC-MS: *m*/*z* 385 (22%), *m*/*z* 200 (98%), *m*/*z* 198 (100%), *m*/*z* 172 (16%), *m*/*z* 170 (16%). ESI-MS(+) C_13_H_11_Br_2_N_3_NaO: *m*/*z* 405.9156.

NMR ^1^H (DMSO-d6, 400 MHz, δ): 10.21 (s, 1H, NH), 7.99 (s, 1H, NH), 7.80 (d, 1H, *J* = 2.3 Hz), 7.31–7.28 (l, 3H), 6.74–6.70 (l, 3H), 6.54 (s, 2H, NH_2_).

#### 3.2.4. Synthesis of 2-amino-N′-(m-tolyl)benzohydrazide (**1i**)

Pale solid, 88% yield. GC-MS: *m*/*z* 241 (31%), *m*/*z* 121 (11%), *m*/*z* 120 (100%), *m*/*z* 92 (21%). ESI-MS(+) C_14_H_15_N_3_NaO: *m*/*z* 264.1109.

NMR ^1^H (MeOD, 400 MHz, δ): 7.58 (d, 1H, *J* = 11.9 Hz), 7.22 (t, 1H, *J* = 7.8 Hz), 7.06 (t, 1H, *J* = 7.8 Hz), 6.78 (d, 1H, *J* = 8.2 Hz), 6.71–6.63 (m, 4H), 2.26 (s, 3H).

#### 3.2.5. Synthesis of 2-amino-N′-(p-tolyl)benzohydrazide (**1j**)

Pale solid, 88% yield. GC-MS: *m*/*z* 241 (34%), *m*/*z* 121 (11%), *m*/*z* 120 (100%), *m*/*z* 106 (7%), *m*/*z* 92 (20%). ESI-MS(+) C_14_H_15_N_3_NaO: *m*/*z* 264.1110.

NMR ^1^H (MeOD, 400 MHz, δ): 7.57 (d, 1H, *J* = 7.9 Hz), 7.22 (t, 1H, *J* = 7.7 Hz), 7.01 (d, 2H, *J* = 8.0 Hz), 6.80–6.77 (m, 3H), 6.66 (t, 1H, *J* = 7.9 Hz), 2.23 (s, 3H).

#### 3.2.6. Synthesis of 2-methyl-3-(phenylamino)quinazolin-4(3H)-one (**2a**)

Pale solid, 85% yield. GC-MS: *m*/*z* 251 (100%), *m*/*z* 236 (13%), *m*/*z* 146 (30%), *m*/*z* 117 (13%).

#### 3.2.7. 3-((4-bromophenyl)amino)-2-methylquinazolin-4(3H)-one (**2b**)

White solid, 51% yield. GC-MS: *m*/*z* 331 (97%), *m*/*z* 329 (100%), *m*/*z* 172 (41%), *m*/*z* 170 (42%), *m*/*z* 146 (65%).

#### 3.2.8. Synthesis of 6-bromo-3-((4-bromophenyl)amino)-2-methylquinazolin-4(3H)-one (**2c**)

Yellow solid, 79% yield. GC-MS: *m*/*z* 411 (50%), *m*/*z* 409 (100%), *m*/*z* 172 (29%), *m*/*z* 170 (34%).

### 3.3. Promastigotes Test

*L. amazonensis* (MHOM/BR/1973/M2269 strain) was used in this study. Promastigotes were grown in an M199 medium supplemented with 10% fetal bovine serum, 5 ppm of hemin, 50 μg/mL streptomycin, and 100 U penicillin until they reached the stationary phase. Initially, 5.0 × 10^5^ cells were incubated with or without the compounds at variable concentrations. Amphotericin B was used as a positive control. The effect was available 24 h, 48 h, and 72 h after incubation with test compounds. The test culture was washed twice with 1 mL of Hanks Balanced Salt Solution (GIBCO^®^ HBSS). After that, the cultures were incubated overnight with dye MTT added to the culture medium at a final concentration of 0.5 mg/mL. The formazan crystals were dissolved with 200 μL of DMSO, and the formazan concentration was determined using the Epoch 2 Microplate Spectrophotometer (Biotech, Instruments, Winooski, VT, USA) at 570 nm. IC_50_ values were estimated using a sigmoidal model (Log IC_50_) in a normalized variable slope in the GraphPad Prism software (version 9.5 for Windows, San Diego, CA, USA). Two independent assays were performed in triplicate [32].

### 3.4. Arginase Assay

For the compound test against *L. amazonensis* arginase (ARG-L), the recombinant was obtained by overexpressing the enzyme in *Escherichia coli,* followed by purification as described previously [33]. Briefly, an *E. coli* culture containing the gene from *L. amazonensis* arginase culture was grown until DO_600_ = 0.6. Then, the enzyme production was induced with 1 mM IPTG in 1 L of SOB medium enriched with 10 mM of MnSO_4_. After 3 h of induction, the enzyme was purified [33]. An inhibition assay was conducted by incubating the arginase with the test compounds at 100 uM using 50 mM CHES buffer and 50 mM L-arginine at pH 9.5 for 15 min [14]. The negative control was performed without the inhibitor, and quercetin was used as a positive inhibition control [14]. The arginase activity was then measured by quantifying urea by the Berthelot method [34].

### 3.5. Target Fishing with Chemogenomics

To predict the compound targets, a chemogenomics approach was used. First, the chemical structures of the compounds were drawn in ChemDrawn v.12.0.2 and prepared with ChemAxon Standardizer (v.16.5.2.0, 2016, http://www.chemaxon.com, accessed on 30 May 2023), in which the SMILE (Simplified Molecular Input Line Entry Specification) notation was generated and standardized, i.e., aromaticity, charges, and tautomers were assigned. Target predictions were created with Pidgin software v.2 [35,36] using predict_binary.py with 0.5 predicted true positive rate. The software compared the structural similarity between our compounds and molecules using Morgan fingerprints with the biological activity described in the databases and included in the predictive models’ construction. Thus, the molecules present in the database with known molecular targets were compared with the present work, and the results were filtered to obtain only the targets with similarity above the cutoff value. Pteridine reductase 1, from Leishmania major, was selected to perform a search for homologous proteins on the NCBI server using Blastp [37]. The alignment of the amino acid sequences was performed on the Praline server [38]. The sequence and structure of Pteridine reductase 1, from *L. amazonensis*, were obtained in Uniprot under the code O09352.

### 3.6. Molecular Docking Studies

Since there are no available experimental structures of *L. amazonensis* PTR1 in the databases, the AlphaFold [39,40] server was used to obtain its three-dimensional structure. To validate the parameters for docking studies, a redocking approach was applied with GOLD [41] (Genetic Optimization for Ligand Docking) v. 2022.3.0 with the 3D structure of PTR1 from *L. major*, PDB code 1E7W [30], with a resolution of 1.75 Å. The water molecules and cofactors interacting with the inhibitor methotrexate (MTX) were maintained. In this study, all docking-scoring functions with a search radius of 10 and 20 Å related to amino acid residues S111, Y194, and Y191 as references were used to estimate the best parameters. Thus, the goldscore function with 10 Å of search radius around Y191 with flexible residues F113 and Y194 reproduced the best superposition between co-crystallized methotrexate (MTX) and the top docking solution. Twenty runs of the genetic algorithm were performed for each molecule. The cofactor coordinate in the *L. amazonensis* PTR1 structure was obtained by structural alignment with the *L. major* PTR1 crystal structure. Model validation metrics were obtained from the Ramachandran plot on the Procheck server [42] and the Z-score and local energy quality on the Prosa-web server (Appendix A) [43,44].

The inhibitors were built with Avogadro v. 1.93.0 [45], and the energy was minimized using the PM3 semi-empirical method available in the Gaussian 09 package. All docking poses and interaction analyses were performed using Pymol v. 2.4.0a0 [46] and Discovery Studio 2016 Visualizer [47].

## 4. Conclusions

The new anthranyl phenylhydrazide showed better activity against *L. amazonensis* promastigotes when compared to α-α-difluorohydrazides. Compound **1g** (IC_50_ = 1.3 μM) was 10-fold more potent than compound **4** (IC_50_ = 12.7 μM), an α-α-difluorohydrazide that was characterized previously [16]. The bromine at positions R1 and R4 is better than only bromine at R1 (**1b**) or only at R4 (**1d**). Target fishing for anthranyl phenylhydrazides indicates the possible mechanism of action for these compounds by inhibiting the parasite enzyme PTR1. The scaffold of anthranyl phenylhydrazide of series 1 is essential for antileishmanial activity. Thus, we conclude that these compounds are promising for developing new substances against *Leishmania*.

## Figures and Tables

**Figure 1 pharmaceuticals-16-01120-f001:**
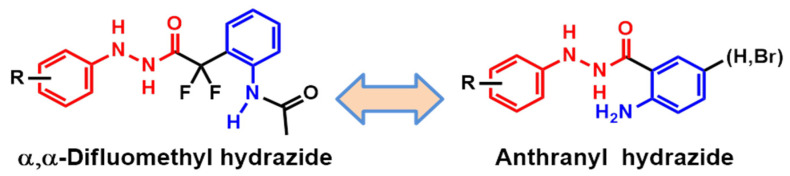
Anthranyl hydrazides designed from α, α-difluoromethylhydrazide.

**Figure 2 pharmaceuticals-16-01120-f002:**
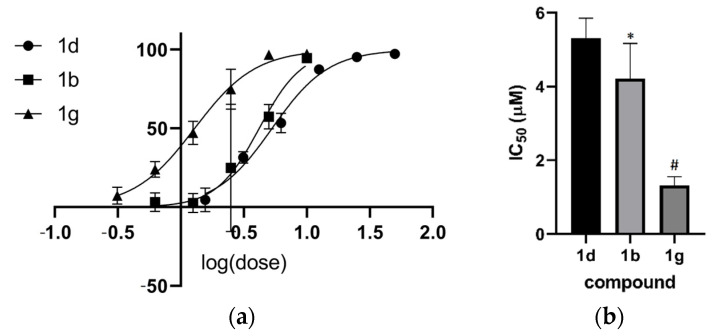
Comparison of promastigote activity of the anthranyl hydrazides series. (**a**) Curve log (dose) response; (**b**) IC_50_ comparison. # *p* < 0.001 vs. **1d** and **1b**; * *p* < 0.05 vs. **1d**.

**Figure 3 pharmaceuticals-16-01120-f003:**
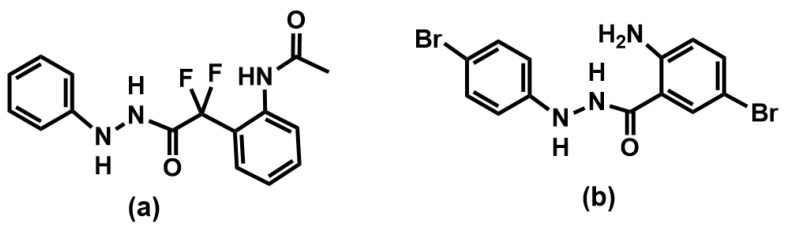
The structure of α-α-difluorohydrazides and anthranyl phenylhydrazide: (**a**) compound **4** [16] and (**b**) compound **1g**.

**Figure 4 pharmaceuticals-16-01120-f004:**
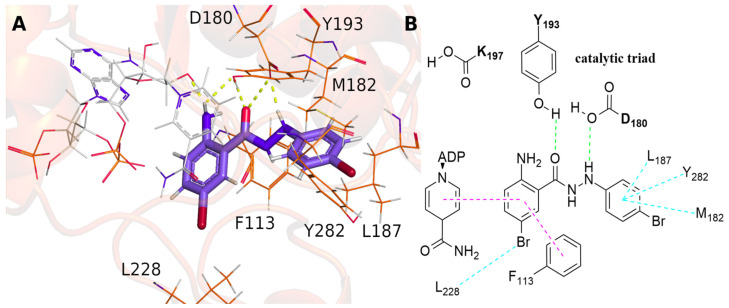
(**A**) Putative binding pose of inhibitor 1g with PTR1 as suggested by molecular docking. (**B**) Scheme with the main interactions carried out in the active site. Hydrogen bond, π-stacked, and alkyl/π-alkyl interactions are represented by green, magenta, and cyan dashed lines, respectively.

**Table 1 pharmaceuticals-16-01120-t001:** Screening of compounds and comparison of IC_50_ of compounds showed more significant promastigote growth inhibition at 10 μM.

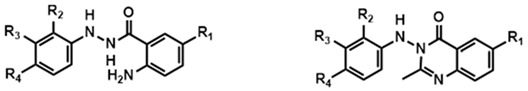
**Compound 1**	**Compound 2**
	**R_1_**	**R_2_**	**R_3_**	**R_4_**	**Inhibition** **at 10 μM (%)**	**IC_50_** **(μM)** **(95% CI)**
**1a**	H	H	H	H	41.3	>10
**1b**	Br	H	H	H	67.7	4.2 (3.460–5.038)
**1c**	H	Br	H	H	30.0	>10
**1d**	H	H	H	Br	55.2	5.3 (4.963–5.670)
**1e**	H	H	H	Cl	59.2	<10
**1f**	H	H	H	F	12.8	>10
**1g**	Br	H	H	Br	97.9	1.3 (1.177–1.411)
**1h**	H	CH_3_	H	H	4.4	>10
**1i**	H	H	CH_3_	H	nd	~20
**1j**	H	H	H	CH_3_	nd	~20
**2a**	H	H	H	H	inactive	-
**2b**	H	H	H	Br	inactive	-
**2c**	Br	H	H	Br	inactive	-

CI, confidence interval; nd, not determined.

## Data Availability

Data is contained within the article and Appendix A.

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
