# Peer review of "Design and Synthesis of New Anthranyl Phenylhydrazides: Antileishmanial Activity and Structure–Activity Relationship"

_pharmaceuticals, 2023, doi:10.3390/ph16081120_

Round 1

Reviewer 1 Report

Accept in present form after Minor editing of English language required

Accept in present form after Minor editing of English language required

Author Response

Response to Reviewer 1 Comments

Point 1: Accept in present form after Minor editing of English language required

Response 1: The authors thank the review suggestion. The manuscript was improved by English editing.

Reviewer 2 Report

Manuscript ID: pharmaceuticals-2506172

Title of the manuscript: Design and synthesis of new anthranyl phenylhydrazides: antileishmanial activity and 2 structure-activity relationship

The current research manuscript describes the Design and synthesis of new anthranyl phenylhydrazides: antileishmanial activity and structure-activity relationship. In the reviewer’s opinion, the manuscript has been planned appropriately but required rigorous corrections. The reviewer also feels that a potential reader can easily understand the details by consulting the main data outlined in the current version of the manuscript with few exceptions. The write-up requires revision to some extent and to the point which is crucial for a acceptance of the research article. In the current version, the reviewer is left with the feeling that despite the significant efforts to conduct research work; this work cannot advance for publication in its present form. I would recommend publishing this work after major modifications and addressing the raised concerns as follows:

Comment 1. The title include the synthesis of Anthranyl phenylhydrazides. Why the word “anthranyl” is being used everywhere if it is not containing anthracene moiety. The probable name of your skeleton is “2-amino-N'-phenylbenzohydrazide”.

Comment 2.  It is clearly not linking properly that the proposed Anthranyl phenylhydrazides are linking with the previously prepared α, α-difluoromethylhydrazide. Show “Drug Design” of the same in the manuscript with appropriate references.

Comment 3. Although a plausible number of analogs with varying substitutions have been synthesized however the authors have detailed insufficient discussion on the Chemistry part. Also elaborate the SAR studies. Authors are advised to prepare analog containing Electron- withdrawing group (NO2, F etc.) and then include in SAR studies as only analogues with Electron-donating groups have been prepared. Possible effects of Cl, Br, Me, NO2 and other functional groups should also be provided in the SAR section.

Comment 4. Redraw Figure 3

Comment 5. Line 110-111 looks speculative and vague. It requires complete explanation with proper references. Similarly, line 177-178 also looks speculative. Change it appropriately with proper reference(s).

Comment 6: 13C NMR data of all the synthesized compounds are not given. Authors should report the 13C NMR spectra of each unknown compounds. The supporting information should also include the 1H NMR as well as 13C NMR spectral data of all the synthesized compounds.

Comment 7: Only 13 compounds have been prepared for this study. Authors should prepare more compounds to develop proper SAR studies.

Manuscript ID: pharmaceuticals-2506172

Title of the manuscript: Design and synthesis of new anthranyl phenylhydrazides: antileishmanial activity and 2 structure-activity relationship

The authors have written a moderate level of English language which requires further improvements.

Author Response

Response to Reviewer 2 Comments

The current research manuscript describes the Design and synthesis of new anthranyl phenylhydrazides: antileishmanial activity and structure-activity relationship. In the reviewer’s opinion, the manuscript has been planned appropriately but required rigorous corrections. The reviewer also feels that a potential reader can easily understand the details by consulting the main data outlined in the current version of the manuscript with few exceptions. The write-up requires revision to some extent and to the point which is crucial for a acceptance of the research article. In the current version, the reviewer is left with the feeling that despite the significant efforts to conduct research work; this work cannot advance for publication in its present form. I would recommend publishing this work after major modifications and addressing the raised concerns as follows:

Comment 1. The title include the synthesis of Anthranyl phenylhydrazides. Why the word “anthranyl” is being used everywhere if it is not containing anthracene moiety. The probable name of your skeleton is “2-amino-N'-phenylbenzohydrazide”.

Response 1: The word anthranyl is being used because hydrazide contains a moiety derived from anthranilic acid.

Comment 2.  It is clearly not linking properly that the proposed Anthranyl phenylhydrazides are linking with the previously prepared α, α-difluoromethylhydrazide. Show “Drug Design” of the same in the manuscript with appropriate references.

Response 2: The previously prepared α,α-difluoromethylhydrazide served as a prototype for the design of anthranyl phenylhydrazides using molecular simplification.

Comment 3. Although a plausible number of analogs with varying substitutions have been synthesized however the authors have detailed insufficient discussion on the Chemistry part. Also elaborate the SAR studies. Authors are advised to prepare analog containing Electron- withdrawing group (NO2, F etc.) and then include in SAR studies as only analogues with Electron-donating groups have been prepared. Possible effects of Cl, Br, Me, NO2 and other functional groups should also be provided in the SAR section.

Response 3: The study was carried out with substituents that have different electronic patterns, such as methyl, which is a weak electron donor and is even an activator group for electrophilic substitution reactions in aromatic rings, and halogens (F, Cl, and Br) are weak deactivators, they donate electrons to the aromatic ring by resonance but withdraw electrons by inductive effect, and this is the predominant effect.

Comment 4. Redraw Figure 3

Response 4: Figure 3 was redrawn and inserted into the manuscript.

Comment 5. Line 110-111 looks speculative and vague. It requires complete explanation with proper references. Similarly, line 177-178 also looks speculative. Change it appropriately with proper reference(s).

Response 5: We agree with the reviewer and remove the speculative phrase from lines 110-111 and justify better the discussion in lines 177-178 (lines 189-195 in the revision version) as follows:

revision lines 125-129 (old 110-111)

Leishmania spp. use several approaches for resistance against drugs [3]. One is ABC transporter MRPA (PGPA) in antimony resistance described for Leishmania infantum [19]. The PRP1, an ABC transporter superfamily member, was correlated with pentamidine resistance in Leishmania major [20]. A new study must be performed to understand why the hydrazide compounds were inactive against intracellular amastigotes.  

lines 177-178 were replaced (206-211):

Further studies need to be conducted to validate the molecular docking hypothesis into PTR1 inhibition, such as enzymatic binding assays. However, PTR1 is a recognized target for leishmanicidal drug design [27,31] since it provides enough folate to guarantee the parasite’s survival [32]. Taken together, this preliminary study, associated with the inhibition in promastigote cells and the prediction by chemogenomics, is a promising molecular target for developing new inhibitors based on the phenylhydrazide scaffold.

Comment 6: 13C NMR data of all the synthesized compounds are not given. Authors should report the 13C NMR spectra of each unknown compounds. The supporting information should also include the 1H NMR as well as 13C NMR spectral data of all the synthesized compounds.

Response 6: The ¹³C NMR spectrum was not performed because all synthesized compounds are known in the literature. ¹H NMR, GC-MS, and high-resolution MS were performed for all hydrazides to corroborate their structure and purity. For quinazolones that were inactive, GC-MS was performed to verify the purity and mass consistent with the product.

Comment 7: Only 13 compounds have been prepared for this study. Authors should prepare more compounds to develop proper SAR studies.

Response 7: Based on our positive results, we will expand the number of compounds in a future study.

Comment 8: The authors have written a moderate level of English language which requires further improvements.

Response 8: The manuscript was improved by English editing.

Reviewer 3 Report

Maquiaveli et al., present a study for the characterization of phenylhidrazines as putative Leishmanicidal agents. I found the work technically sound for the most part. Here are some comments/suggestions for the authors:

The introductions offers a good summary, however I think that some information can be added on the subject of hydrazine compounds and their role as therapeutic compounds.

On a similar subject, the use of brominated compounds is often regarded as a divisive subject. I must say that I have no problem with bromine; yet I do think that this matter should be acknowledged by the authors and provide some commentary.

The computational protocol is lacking in several details. For instance the parameters for the construction of the homology model. Also further validation is needed beyond Ramachandran plot and the predicted error showed in Figure S4.

Similarly, the chemogenomics scheme should be expanded upon and provide more context on the methodology.

Discussion of docking results is good, but I think that more information on PTR1 is needed. Also, the discussion on structural features of the binding site should be improved.

The authors should consider the inclusion of molecular dynamics simulation for both homology model validation and the assessment of the identified binding modes.

Finally, a minor detail is the low resolution of Figure S2, the structures are barely visible.

Minor revision is needed to improve the manuscript, as there are some grammatical errors in it.

Author Response

Response to Reviewer 1 Comments

Maquiaveli et al., present a study for the characterization of phenylhidrazines as putative Leishmanicidal agents. I found the work technically sound for the most part. Here are some comments/suggestions for the authors:

Point 1: The introductions offers a good summary, however I think that some information can be added on the subject of hydrazine compounds and their role as therapeutic compounds.

Response 1: To the best of our knowledge, there are no hydrazides with leishmanicidal activity described in the literature, except for the hydrazides previously published by our group, whose structure was based on eflornithine, an inhibitor of the enzyme ornithine decarboxylase (ODC).

We improved the introduction by adding the following text:

"Until now, no other hydrazides with leishmanicidal activity have been described in the literature, except for those previously published by our group [16], for which its structure was based on eflornithine, an inhibitor of the ODC enzyme. This study tests 13 structural hydrazides related to the previously prepared α,α-difluoromethylhydrazide as a prototype for designing anthranyl phenylhydrazides using molecular simplification. These anthranyl phenylhydrazides showed better activity against Leishmania amazonensis promastigotes than the previous α,α-difluoromethylhydrazide series [17]. "

Point 2: On a similar subject, the use of brominated compounds is often regarded as a divisive subject. I must say that I have no problem with bromine; yet I do think that this matter should be acknowledged by the authors and provide some commentary.

Response 2: Indeed, brominated compounds are a controversial topic within medicinal chemistry, but our best results were with compounds containing bromine in their structure and we cannot neglect this until we have all the answers about their activity. Furthermore, there are known drugs that have bromine in their structure.

Point 3: The computational protocol is lacking in several details. For instance the parameters for the construction of the homology model. Also further validation is needed beyond Ramachandran plot and the predicted error showed in Figure S4.

Response 3: As suggested by the reviewers, an additional validation was performed on the Prosa-web server (Figure S6). However, it is not possible to provide the three-dimensional model construction parameters, since it was obtained directly from the AlphaFold database, under the UniProt code O09352. The choice to obtain the model directly from the AlphaFold database instead of building it by comparative modeling was due to the fact that it was the highest one ranked in CASP14.

Figure S6: (A) Comparison of the z-score of our model against all protein chains in PDB determined by X-ray crystallography (light blue) or NMR spectroscopy (dark blue). (B) Local model quality was observed by residue energies averaged in two windows size 10 (light green) and 40 (green).

Point 4: Similarly, the chemogenomics scheme should be expanded upon and provide more context on the methodology.

Response 4: This chemogenomic scheme was improved as follows (lines 287-301):

"To predict the compound targets, a chemogenomics approach was used. First, the chemical structures of the compounds were drawn in ChemDrawn v.12.0.2 and prepared with ChemAxon Standardizer (v.16.5.2.0, 2016, http://www.chemaxon.com), in which the SMILE (Simplified Molecular Input Line Entry Specification) notation was generated and standardized, i.e., aromaticity, charges, and tautomers were assigned. Target predictions were created with Pidgin software v. 2 [37,38] using predict_binary.py with 0.5 predicted true positive rate. The software compared the structural similarity between our compounds and molecules using Morgan fingerprints with biological activity described in the databases included in the predictive models’ construction. Thus, the molecules present in the database with known molecular targets were compared with the present work, and the results filtered to obtain only the targets with similarity above the cutoff value. Pteridine reductase 1, from Leishmania major, was selected to perform a search for homologous proteins on the NCBI server using Blastp [39]. Alignment of amino acid sequences was performed on the Praline server [40]. The sequence and structure of Pteridine reductase 1, from L. amazonensis, were obtained in Uniprot under the code O09352."

Point 5: Discussion of docking results is good, but I think that more information on PTR1 is needed. Also, the discussion on structural features of the binding site should be improved.

Response 5: Lines 172-198 enclosed the final version of the improved text as follows:

"Pteridine reductase is a NADPH-dependent reductase that catalyzes the reduction of folate and biopterin into their biologically active forms, tetrahydrofolate and tetrahydrobiopterin, respectively[13]. It is the only enzyme to perform this reduction in Leishmania, proving essential for their in vivo growth by gene knockout studies[14]. Besides that, Leishmania is auxotrophic for folate and other pterins required in critical pathways related to the synthesis of nucleic acids and proteins [15], [16]. Therefore, the drug development of new inhibitors for PTR1 would be a promising strategy for the treatment of leishmaniasis [17]–[19]. Although target fishing provides significant results on possible molecular targets, this technique depends on molecular scaffolds in the database, often with sparse data for specific organisms. Thus, molecular docking was performed to obtain the theoretical binding mode of the designed compounds with PTR1 from L. amazonensis. Despite the binding site of PTR1, from L. amazonensis, being very similar to L. major  (RMSD of 1.30 Å over 264 residues), 24 amino acids are not conserved (Figure S1). Of these 24 residues, only two residues, A229 and Y240, are within 5 Å of distance from the inhibitor methotrexate. The catalytic center is formed by β1, β4, β5 and β6, located in the C-terminal regions, α1 and α5 of the N-terminal and a loop between β6 and α66. The results showed that compound 1g could interact with two residues of the catalytic triad (Y193 and D180) by hydrogen bonding. Besides that, π-π stacked interactions with F113 and the cofactor were also observed (Figure 4). Notably, the other molecules, in general, presented a similar putative binding mode (Figure S2), indicating that this molecular scaffold may be relevant to the drug design of new candidates to treat leishmaniasis. Similarity with the crystallized co-inhibitor methotrexate could also be observed (Figure S3), mainly concerning the interaction with the catalytic residue Y193 (Y194 in PTR1 from L. major). Gourley et al. described the mechanism of action of PTR1, involving hydrogen bonds between Y194 and a nucleophilic center of the substrate, bound in such a way that the transition state is stabilized to provide the proton transferred in the reaction [20]. Therefore, altogether, the observed interactions support the hypothesis of PTR1 as a target of the studied compounds. "

Point 6: The authors should consider the inclusion of molecular dynamics simulation for both homology model validation and the assessment of the identified binding modes.

Response 6: We believe that molecular dynamics calculations, despite providing relevant information on conformation and stability of the ligand in the protein cavity, would be outside the scope of the present work. Additional efforts are being made to validate PTR1 as the molecular target of the synthesized molecules and once we have experimental confirmations, molecular dynamics calculations as well as binding energy calculations will be performed.

Point 7: Finally, a minor detail is the low resolution of Figure S2, the structures are barely visible.

Response 7: The resolution of the figure has been changed as suggested. Figure S2 was divided into two figures to fit on the page.

Point 8: Minor revision is needed to improve the manuscript, as there are some grammatical errors in it.

Response 8: The manuscript was improved by English editing.

Round 2

Reviewer 3 Report

The authors have made significant editions to the manuscript. Most of my concerns have been adressed.

Just as a final comment, in the previous round my suggestion on the presence of bromine atoms was aimed for the addition of said comment/citation on the matter in the main text.

Minor editing/style errors persist